# Hypoxia-Nitric Oxide Axis and the Associated Damage Molecular Pattern in Cutaneous Melanoma

**DOI:** 10.3390/jpm12101646

**Published:** 2022-10-04

**Authors:** Corina Daniela Ene, Ilinca Nicolae

**Affiliations:** 1“Carol Davila” Nephrology Hospital, 010731 Bucharest, Romania; 2Faculty of General Medicine, “Carol Davila” University of Medicine and Pharmacy, 020021 Bucharest, Romania; 3Dermatology Department, “Victor Babes” Clinical Hospital for Infectious Diseases, 030303 Bucharest, Romania

**Keywords:** hypoxia, nitric oxide, malignant cutaneous melanoma, metabolic reprogramming, therapeutic targets

## Abstract

Hypoxia was intensively studied in cancer during the last few decades, being considered a characteristic of the tumor microenvironment. The aim of the study was to evaluate the capacity of tumor cells to adapt to the stress generated by limited oxygen tissue in cutaneous melanoma. We developed a case–control prospective study that included 52 patients with cutaneous melanoma and 35 healthy subjects. We focused on identifying and monitoring hypoxia, the dynamic of nitric oxide (NO) serum metabolites and posttranslational metabolic disorders induced by NO signaling according to the clinical, biological and tumoral characteristics of the melanoma patients. Our study showed high levels of hypoxia-inducible factor-1a (HIF-1a) and hypoxia-inducible factor-2a (HIF-2a) in the melanoma patients. Hypoxia-inducible factors (HIFs) control the capacity of tumor cells to adapt to low levels of oxygen. Hypoxia regulated the nitric oxide synthase (NOS) expression and activity. In the cutaneous melanoma patients, disorders in NO metabolism were detected. The serum levels of the NO metabolites were significantly higher in the melanoma patients. NO signaling influenced the tumor microenvironment by modulating tumoral proliferation and sustaining immune suppression. Maintaining NO homeostasis in the hypoxic tumoral microenvironment could be considered a future therapeutic target in cutaneous melanoma.

## 1. Introduction

Hypoxia was intensively studied during the last few decades, being considered a universal stimulus for many transcriptional programs and a coordinator for multiple biological processes [1,2]. Some studies showed that intratumor hypoxia influences treatment efficacy and prognoses [3]. The stress generated by limited tissue oxygen induces an adapted response to the energetic needs [3]. Activation of hypoxia-inducible factors (HIFs) is an important response of cells at low oxygen concentrations. Three heterodimeric factors (HIF-1, HIF-2, and HIF-3) composed of oxygen-sensible alpha units and stable beta units, are described in the literature [4,5]. In physiological conditions, HIF-1a and HIF-2a are proteasomally degraded. In hypoxic conditions, HIF-1a and HIF-2a are directed in the cellular nucleus and dimerize with HIF-1β. HIF-1 and HIF-2 are expressed in many tissues and regulate the expression of more than 1000 genes involved in hypoxia [4,5,6,7,8,9,10,11].

The human skin is hypoxic due to the absence of vasculature, though many hypoxia markers are highly expressed in the skin [11]. Melanocytes in the dermo-epidermic zone live and develop in hypoxic conditions. The hypoxic microenvironment of the skin could favor the oncogenic transformation of melanocytes [11,12]. Hypoxia could influence cutaneous melanoma progression by evidence of a proliferative phenotype and invasive potential [13]. The tumoral hypoxic microenvironment is responsible for the downregulation of melanocyte differentiation, decrease in melanocyte marker expression (MITF, MLANA, and WNT5A), overexpression of genetic and gained heterogenicity, formation and release of exosomes for tumor cells, angiogenesis, metastasis, and chemotherapy resistance [1,2,6,14,15,16,17,18,19]. The hypoxia level was proven to be a prognosis factor for overall survival and a predictor of the immune microenvironment. In the last decade, the hypoxic microenvironment was considered an important pathogenicity element in solid tumors [14,18,20,21].

An important mediator of tumor hypoxia is nitric oxide (NO). The molecular mechanisms induced by hypoxia in the tumor microenvironment by changing the NO cycle are not fully understood. HIF-1 and HIF-2 promote the angiogenic properties of tumor-associated macrophages (TAMs), which accumulate in hypoxic tissues. Moreover, HIF-1a regulates the function of myeloid-derived suppressor cells (MDSC) in the hypoxic microenvironment by stimulating angiogenesis and NO production [22,23]. In human tissues, two different pathways of NO synthesis are known, with different effects in HIF signaling. First, endogen NO synthesis from an L-arginine and oxygen reaction (Equation (1)) is catalyzed by nitric-oxide synthase (NOS) as follows:2L − ARG + 3NADPH + 3H(+) + 4O_2_ + 2NO + 2L − CITRULLINE + 3NADP(+) + 4H_2_O(1)

Arginine NO-dependent synthesis through an oxidative pathway is modulated by the tissue’s oxygen concentration, NG-dimethyl L-Arg (ADMA), NG-methyl-L Arg (L-NMA), calcium, NOS isotype distribution, and activity (constitutive or inducible) [24,25,26,27,28,29,30].

The second pathway of NO synthesis (Equation (2)) is NOS-independent and derived from dietary nitrates and nitrites, from an NOS-mediated endogen diet, and commensal microorganisms’ activity. The reaction is controlled by mitochondrial cytochrome oxidase as follows:FE(2+) + O_2_(−) + H(+) + FE(3+) + NO + OH(−)(2)

NOS-independent NO synthesis, known as the reductive pathway, is regulated by the COX4-2 subunit. Oxidative pathways (Equation (1)) are active in normoxia, while the reductive one (Equation (2)) is functional in normoxia and hypoxia [27,28,29,30]. HIF-NO signaling was studied in degenerative disease, ischemia-perfusion, cancers, inflammation, autophagy, apoptosis, cells survival, immunity, and angiogenesis [31]. The HIF-NO relation is complex, with oncogenic and suppressor effects in cancer cells [31]. Many recent studies showed that hypoxic cells can adapt to low oxygen levels by changing the transcriptional and translational responses [32].

Melanocytes express all NOS isoforms (neuronal, endothelial, and inducible). NO metabolites induce a signaling cascade concreted in melanogenesis alteration and protein post-translational changes. All these are based on nitration reactions, S-Nitrosilation, oxidation, and protein thiol reduction. These post-translational changes alter DNA–protein interactions, protein thiols, and redox-sensible protein functions [14,20,32,33,34]. The NO cycle induces the formation of toxic compounds (NO_2_ and ONOO−) involved in lipid peroxidation [33,35,36]. NO-dependent lipid peroxidation is influenced by peroxide free radicals’ concentrations [37]. The tumor cells have different needs of NO associated with the antitumor potential [38,39]. Much research has shown that NO could damage DNA by inducing mutagenicity and cytotoxicity [37,38]. Through its oxidative and disseminating effects, NO inhibits the repair of DNA lesions and induces p53 mutagenesis [17]. NO also inhibits the synthesis of DNA by hypoxia or iron removal [38].

Recently, some researchers proved that HIF synthesis is an early stimulus for the proteins and enzymes involved in NO production and in metabolic reprogramming of cancer cells [3,12]. We developed a case control prospective study for evaluating the hypoxia-NO relation in cutaneous melanoma focused on hypoxia monitoring, NO serum metabolite dynamics, and the post-translational changes induced by NO signaling. The results of the present study report the tumor, clinical and biological characteristics of patients with cutaneous melanoma. Identifying the NO equilibrium is very important in cancer biology for developing future therapeutic strategies that modulate NO homeostasis and redox signaling.

## 2. Materials and Methods

### 2.1. Study Participants

We focused on the identification and monitoring of hypoxia, the dynamics of nitric oxide (NO) serum metabolites, and post-translational metabolic disorders induced by NO signaling according to the clinical, biological, and tumoral characteristics of melanoma patients.

The authors developed a case–control prospective study over a period of two years and included two groups: a melanoma group with 52 subjects and a control group with 35 healthy subjects. All the patients signed informed consent forms, and all the procedures were performed according to the Declaration from Helsinki from 1975. Patients were selected from those who attended the Carol Davila Clinical Hospital of Nephrology and Victor Babes Hospital, and the study protocol was approved by the Ethics Committee of Victor Babes Hospital. The patients with malignant cutaneous melanoma were diagnosed by a histological exam after surgical removal of the tumor. All the patients were evaluated for their TNM stages by computed tomography or nuclear magnetic resonance. The tumor characteristics registered were as follows: tumor localization, histopathological type, Breslow index, Clark level, the presence or absence of ulceration, and TNM stage. The patients included in the melanoma group were over 18 years old and with different stages of melanoma without any treatment regimen before clinical or biological data were registered. In the control group, we enrolled healthy subjects over 18 years old with adequate nutritional statuses. The exclusion criteria were tobacco use, drug abuse, alcoholism, the use of anti-inflammatory therapy (corticoids or nonsteroidal anti-inflammatories), vitamin or other supplement use, and pregnancy.

### 2.2. Laboratory Data

Blood samples were collected from all the study participants after 12 h of fasting using a holder-vacutainer system. Centrifugation of the blood samples was conducted at 3000× *g* for 10 min after 1 hour of keeping the blood at room temperature. The serum was separated and frozen at −80 degrees before analysis. We excluded the hemolyzed, icteric, lactescent, or microbiologically contaminated samples. The samples for laboratory determination were collected from patients in the healthy group after signing an informed consent form, and in the melanoma group, this was performed after the histopathological diagnosis of the tumor and its staging.

Hypoxia was evaluated by the serum levels of HIF-1a and HIF-2a using a sandwich ELISA Kit (MyBioSource, USA), a solid phase sandwich enzyme-linked immunosorbent assay. Both HIF-1a antibodies and HIF-2a antibodies, were coated onto the microwells. After incubation with the patient’s serum, the HIF protein was captured by the coated antibody. Following extensive washing, an HIF detection antibody was added to detect the captured HIF protein. An HRP substrate, TMB, was added to develop the color. The magnitude of the optical density for this developed color was proportional to the quantity of HIF-1a with respect to the HIF-2a proteins. The obtained values were expressed in ng/mL of serum.

The serum nitric oxide metabolite characteristics evaluated were the direct nitrite, total nitrite, and nitrate. This assay determined the nitric oxide concentrations based on the enzymatic conversion of nitrate to nitrite by the nitrate reductase. The reaction was followed by colorimetric detection of nitrite with an azo dye product which absorbed light at 540–570 nm. The results were expressed in μmol/L.

Nitration was evaluated by the serum level of nitrotyrosine (3-NT), a specific biomarker for oxidative alterations mediated by a peroxynitrite antioxidant that induced the nitration of the rest of the tyrosine from the proteins. The circulant 3-NT was assessed by immunoenzymatic ELISA, and the results were expressed in μmol/L.

Carbonylation was evaluated by the serum levels of the carbonyl groups, malondialdehyde (MDA), 4-hydroxy-2-nonenal (HNE), and thiobarbituric acid-reactive substances (TBARS.) The carbonyl groups were determined by the spectrophotometric method in a reaction with 2,4-dinythrophenylhydrasine that induced hydrazone formation. The results were expressed in μmol/L, and 4-HNE and MDA were assessed by the competitive ELISA method. The wells were pre-coated with substrate, and the final product colorimetric evaluation was made at 450 nm. The results were expressed in μg/mL of serum for 4-HNE and in ng/mL of serum for MDA. The MDA formed a complex with thiobarbituric acid-reactive substances (TBARS) that was measured using the spectrophotometric method (BS-3000M Semi-Automatic Chemistry Analyzer, Sinnowa, Nanjing, China) and read at a wavelength of 532 nm. The results were expressed in μmol/L of serum.

The TDHPs were determined using the spectrophotometric method. We used sodium borohydride (NaBH4, 10 mM) in order to transform the reducible disulfide into free functional thiol group bonds according to the reaction R-S2-R ‘+ NaBH4 → 2 R-SH + BH3 + Na. We used formaldehyde (10 mM, pH 8.2) to remove the NaBH4, which was in excess in the reaction. The levels of the native thiol (NT) and total thiol (TT) were measured using 5,5′-dithiobis-2-nitrobenzoic acid (DTNB, 10 mM) according to the reaction R-SH + DTNB → R-TNB + TNB. Half of the difference between the TT and NT was considered the disulfide (DS) level. The results were expressed in μmol/L of serum. The disulfide/native thiol ratio (DS/NT), disulfide/total thiol ratio (DS/TT), and native thiol/total thiol ratio (NT/TT) were calculated as follows:-The DS/NT was calculated as -S-S-*100/-SH;-The DS/TT was calculated as -S-S-*100/-SH + -S-S-;-The NT/TT was calculated as -SH*100/-SH + -S-S-.

(NT: (-SH); TT: (-SH + -S-S-); DS: (-S-S))

The symmetric dimethylarginine was assessed by the ELISA method’s competitive variant. This method is sensible, reproductible, repeatable, and specific for SDMA with no cross-reactions or interference with other structural analogs. The technique uses a specific primary antibody for unmarked enzymatic SDMA and a secondary antibody specific to the primary antibody which is a marked enzyme. The intensity of the yellow color, measured at 450 nm, was inversely proportional to the SDMA concentration. A high concentration of SDMA decreased the photometric signal. The sample SDMA was calculated based on the standard curve used in identical experimental conditions. The SDMA values were expressed in nmols/L of serum.

The asymmetric dimethylarginine was assessed by the ELISA method’s competitive variant. This method has high sensibility and reproducibility, a large domain of concentration linearity, and the possibility to adapt to varied samples. The intensity of the yellow color, measured at 450 nm, was inversely proportional to the ADMA concentration. The sample ADMA was calculated based on the standard curve used in identical experimental conditions. The ADMA values were expressed in nmols/L of serum.

### 2.3. Statistical Analysis

All the results were analyzed using IBM^®^ SPSS^®^ (Statistics 2015 Version 23.0. IBM Corp, Armonk, NY, USA). We evaluated the normality of the data distribution using the Kolmogorov–Smirnov test. The variation between groups was determined using a parametric test, specifically the *t* test, when two groups were compared or the ANOVA test when more groups were compared, as well as nonparametric tests such as the Mann–Whitney or Wilcoxon tests. The correlation between groups was evaluated using linear regression and the Pearson coefficient, where *p* < 0.05 was considered statistically significant.

## 3. Results

### 3.1. Groups of Characteristics

The characteristics of the subjects included in the study are presented in Table 1. The melanoma and control groups were similar regarding sex, age, and body mass index. When evaluating the skin phototype, we detected a significantly higher the number of patients with the I-II skin phototype in the melanoma group compared with the control group (*p* < 0.05). The serum level of lactate dehydrogenase (LDH) was significantly increased in the melanoma group compared with the control group (*p* < 0.05). For the melanoma group, the tumor characteristics are presented below. Regarding tumor localization, 17.3% of the tumors were head-neck tumors, 42.3% were trunk tumors, and 40.3% were limb tumors. Regarding the histopathological type, 30.7% were nodular, 40.3% were extensive in their surfaces, 13.4% were lenticular, and 15.3% were acral. Regarding the Breslow index, 27% were <1.0 mm, 32.6% were 1.01–2.0 mm, 23% were 2.01–3.0 mm, and 17.3% were >3.01 mm. Regarding the Clark level, 23% were at level II, 32.7% were at level III, 23% were at level IV, and 21.1% were at level V. Ulceration was present in 75% of the evaluated melanoma patients. Regarding the TNM stage, 9.6% were at TNM 0, 13.4% were at TNM 1, 34.7% were at TNM 2, 30.7% were at TNM 3, and 11.5% were at TNM 4.

### 3.2. Levels of Hypoxia in Studied Groups

The markers of hypoxia were significantly higher in the melanoma group, as presented in Table 2. HIF-1a was 1.55-fold higher in the melanoma group compared with the control group (*p* < 0.05), while HIF-2a was 2.32-fold higher in the melanoma patients compared with the control patients (*p* < 0.05). Meanwhile, the HIF1a/HIF2a ratio was 1.5-fold lower in the melanoma group compared with the control group (*p* < 0.05).

### 3.3. NO Metabolism in Studied Groups

The serum nitric oxide metabolite levels in hypoxic conditions are presented in Table 3. The direct nitrite increased 2.24-fold in the melanoma group compared with the control group (*p* < 0.05), the total nitrite increased 2.33-folds in the melanoma group compared with the control group (*p* < 0.05), and nitrate increased 2.41-fold in the melanoma group compared with the control group (*p* < 0.05).

### 3.4. Damage-Related Molecular Pattern in Studied Groups

The damage-related molecular pattern evaluated the nitration, protein carbonylation, thiol-disulphide homeostasis, and methylated arginine, and the results are presented in Table 4. Nitration was overexpressed in the melanoma group, and nitrotyrosine increased 2.92-fold in the melanoma group compared with the control group (*p* < 0.05). Carbonylation also significantly increased in the melanoma group. The PCO had 1.68-fold higher levels in the melanoma group compared with the control group (*p* < 0.001), the 4-HNE had 1.50-fold higher levels in the melanoma group compared with the control group (*p* < 0.001), the TBARS had 1.64-fold higher levels in the melanoma group compared with the control group (*p* < 0.001), and the MDA had 1.81-fold higher levels in the melanoma group compared with the control group (*p* < 0.001).

Thiol-disulphide homeostasis was disrupted in the melanoma group, as presented below. The NT decreased 1.19-fold in the melanoma group compared with the control group (*p* < 0.001), and the TT decreased 1.08-fold in the melanoma group compared with the control group (*p* < 0.001), while the DS increased 1.31-fold in the melanoma group compared with the control group (*p* < 0.001). The DS/NT ratio increased 1.48-fold and the DS/TT ratio increased 1.58-fold in the melanoma group compared with the control group (*p* < 0.001). The NT/TT ratio decreased 1.04-fold in the melanoma group compared with the control group (*p* < 0.001).

The methylated arginine increased significantly, the SDMA increased 1.92-fold, and the ADMA increased 1.48-fold in the melanoma group compared with the control group (*p* ˂ 0.001), while their ratio decreased 1.22-fold compared with the control group (*p* ˂ 0.001).

### 3.5. Hypoxia, NO Metabolism, and Damage-Related Molecular Patterns in Relation to the Breslow Index, Clark Level, and Melanoma Stage

The processes presented above were analyzed in relation to the following tumor characteristics: Breslow index, Clark level, and melanoma stage. The results are presented below.

#### 3.5.1. HIF-1a and HIF-2a Levels in Melanoma According to the Breslow Index, Clark Level, and Melanoma Stage

Regarding the Breslow index, both HIF-1a and HIF-2a had statistically significant increases in the Breslow index, while the HIF-1a/HIF-2a ratio significantly decreased in patients with high Breslow index values (Table 5).

Regarding the Clark level, both HIF-1a and HIF-2a presented statistically significant increased values in the patients, while the HIF-1a/HIF-2a ratio significantly decreased in patients with high Clark levels (Table 6).

Regarding the melanoma stage, both HIF-1a and HIF-2a presented statistically significant higher levels in the patients with advanced melanoma, while the HIF-1a/HIF-2a ratio significantly decreased in patients with high melanoma stages (Table 7).

#### 3.5.2. Serum Nitric Oxide Metabolite Levels in Melanoma According to the Breslow Index, Clark Level, and Melanoma Stage

Regarding the Breslow index, both the direct nitrite and total nitrite presented statistically significant higher levels in the patients with high Breslow index values (Table 8). The nitrate did not vary significantly when comparing the melanoma patients with Breslow indexes of 1.01–2.0 versus <1.0 mm or 2.01–3.0 versus 1.01–2.0. In the melanoma patients with Breslow indexes >3.01, the nitrate presented statistically significant higher levels when compared with melanoma patients with lower Breslow indexes (Table 8).

Regarding the Clark level, both the direct nitrite and total nitrite presented statistically significant increased levels when comparing Clark V with Clark II, III, and IV melanoma patients (Table 9). The nitrate did not vary significantly when comparing Clark level III versus II melanoma patients (Table 9). The nitrate increased significantly in Clark level IV and V melanoma patients when compared with the lower Clark levels (Table 9).

Regarding the melanoma stage, neither the direct nitrite, total nitrite, nor nitrate varied significantly when comparing stage II versus I melanoma patients (Table 10). These parameters were significantly higher when comparing the stage III and IV melanoma patients with the stage II melanoma patients (Table 10).

#### 3.5.3. Nitration and Carbonylation in Melanoma According to the Breslow Index, Clark Level, and Melanoma Stage

Regarding the Breslow index, nitration and carbonylation were intensive processes in melanoma patients with high Breslow indexes (Table 11). Nitrotyrosine was significantly increased with a Breslow index >3.01 when compared with Breslow indexes of <1.0 and 1.01–2.0 or Breslow indexes of 2.01–3.0 and 1.01–2.0 when compared with Breslow indexes <1.0 in melanoma patients (Table 11). The PCO, TBARS, 4-HNE, and MDA showed statistically significant increases in patients with high Breslow indexes (Table 11), but 4-HNE did not vary significantly when comparing Breslow indexes of 2.01–3.0 versus 1.01–2.0 or >3.01 versus 2.01–3.0 (Table 11).

Regarding the Clark level, nitration and carbonylation were intensive processes in melanoma patients with high Clark levels (Table 12). We detected significantly increased serum values of nitrotyrosine and PCO when comparing Clark levels III, IV, and V versus II, levels IV and V versus III, and level V versus IV (Table 12). The 4-HNE had statistically significant higher values when comparing Clark levels III and IV versus II and levels V versus IV and no significant variation when comparing Clark levels IV and V versus III or level V versus IV (Table 12). The TBARS had statistically significant higher values when comparing Clark levels IV versus II and III and levels V versus II and III and no significant variation when comparing Clark level III versus II or level V versus IV (Table 12). The MDA had statistically significant higher levels when comparing Clark levels IV versus II and III and levels V versus II, III, and IV and no significant variation when comparing Clark level III versus II or level V versus IV (Table 12).

Regarding the melanoma stage, the nitrotyrosine had statistically significant increased serum levels when comparing melanoma stags III and IV versus I, stages III and IV versus II, and stage IV versus III and no significant variation when comparing melanoma stage II versus I (Table 13). The PCO and MDA showed statistically significant increases with every melanoma stage (Table 13). The 4-HNE had statistically significant increased serum levels when comparing melanoma stages III and IV versus I and stage IV versus III and no significant variation when comparing melanoma stage II versus I, III versus II, and IV versus III (Table 13). The TBARS had statistically significant increased serum levels when comparing melanoma stages II, III, and IV versus I, stage IV versus II, and stage IV versus III. No significant variation was detected when comparing melanoma stage III versus II (Table 13).

#### 3.5.4. Methylated Arginine Levels in Melanoma According to the Breslow Index, Clark Level, and Melanoma Stage

Regarding the Breslow index, both the SDMA and ADMA levels showed statistically significant increases in the Breslow index, while the ADMA/SDMA ratio had no statistically significant variation (Table 14).

Regarding the Clark level, the SDMA presented statistically significant increased values when comparing Clark levels IV and V versus II and levels IV and V versus III and no statistical variation when comparing Clark level III versus II and level V versus IV (Table 15). The ADMA showed a statistically significant increase with the Clark level, while the ADMA/SDMA ratio had no significant variation (Table 15).

Regarding the melanoma stage, both the SDMA and ADMA presented statistically significant increased levels in the advanced melanoma stages, while the ADMA/SDMA ratio had no statistically significant variation (Table 16).

### 3.6. The Relation between the Tumor Microenvironment, Hypoxia, and Tumor Characteristics

The factors of hypoxia, NO metabolites, carbonylation, TDHP, and arginine methylation were studied in relation to the characteristics of the tumor, and they are presented in Table 17. Regarding the tumors, we determined a strong positive correlation with HIF-1a and the Clark level and Breslow index, as well as statistically significant and no correlation with the LDH and TNM stage and the presence or absence of ulceration, respectively. HIF-2a had a strongly positive correlation only with the LDH and TNM. The NO metabolites, direct nitrate, and nitrate correlated positively and were statistically significant with the LDH, Clark level, Breslow index, and TNM stage but did not correlate with the presence or absence of ulceration.

Meanwhile, 3-nitrotyrosine, a marker of nitrosative stress, carbonylic groups, and TBARS, markers of protein carbonylation, correlated positively with the LDH, Clark level, and Breslow index. 4-HNE correlated positively and the correlation was statistically significant with all tumor characteristics except the presence/absence of ulceration. MDA correlated positively and the correlation was statistically significant with the Clark level, Breslow index, and TNM stage.

The TDPH presented interesting correlations with the tumor characteristics. The NT correlated negatively and was statistically significant with the LDH and Breslow index and positively correlated with the Clark level, while the TT correlated negatively and was statistically significant with the LDH, Breslow index, and Clark level. The DS, DS/NT, DS/TT, and NT/TT had no correlation with the tumor characteristics.

The methylated arginine had different relations with the tumor characteristics: the SDMA correlated positively and was statistically significant with the Clark level and presence or absence of ulceration and negatively at the level of significance with the Breslow index. The ADMA correlated positively and was statistically significant with the LDH, Clark level, Breslow index, and TNM stage. Moreover, their ratios correlated positively and were statistically significant only with the Clark level.

The relations between the HIF-NO axis and damage-related molecular pattern in the melanoma group are presented in Table 18. The NO metabolites were strongly correlated positively with the hypoxia markers with respect to their ratios. All studied nitration and carbonylation factors correlated positively with and were statistically significant for the hypoxia markers and NO metabolites. When we analyzed the thiol group relation to the HIF-NO axis, we found an interesting negative correlation with statistical significance between NT and TT with the HIF-1a, HIF-2a, direct nitrite, and nitrate. In addition, it is important to mention the positive correlation of DS with these markers. Regarding the methylated arginine, the SDMA correlated positively with and was statistically significant for HIF1a, while the ADMA correlated positively with all hypoxia markers and NO degradation products. Their ratio correlated with the HIF2a and HIF1a/HIF2a ratio.

## 4. Discussion

The present study was focused on the molecular pathogeny of cutaneous melanoma, showing its relation with NO production in hypoxic conditions and cell metabolism reprogramming.

In our study, HIF-1a and HIF-2a presented high levels in melanoma patients compared with the control group. The HIF-1a and HIF-2a expressions were different in relation to the tumor characteristics. HIF-1a positively correlated with the Clark level and Breslow index, while HIF-2a varied with the LDH and tumor stage. An interesting relation was the negative correlation between the HIF-1a/HIF-2a ratio and the Clark level, as well as the Breslow index and tumor stage. These data could show that hypoxia is a common characteristic of solid tumors. Moreover, HIF-1a and HIF-2a proteins are involved in melanogenesis and could be involved in cutaneous melanoma evolution. Though HIF-1a and HIF-2a could play an essential role in regulating the oncogenic processes induced by hypoxia, the expressions and activities of these factors are differentially regulated in a hypoxic tumor microenvironment.

HIF-1a and HIF-2a have similar structures but different responses to hypoxia and gene regulation [40,41,42,43]. HIF-1a mediates cell adaptation to acute hypoxia, while HIF-2a does so for chronic hypoxia [40,43]. Some studies showed that HIF-2a stimulates tumorigenesis while HIF-1a acts as a tumor suppressor [42]. HIF-2a is associated with low prognoses in patients with solid tumors [44]. In our study, both HIF-1a and HIF-2a correlated with low melanoma prognostic markers. The function of the proteins overexpressed in cancer, such as CAIX, PDK, BNip3, Mxi-1, VEGF, and EPO, is regulated by HIF-1a in acute hypoxia. The tumor expressions of GLUT-1, Cyclin D1, TGF-α, VEGF, and EPO are specific targets of HIF-2a in chronic hypoxia when an oxidative phenotype is induced and even when tumor aggressivity and treatment resistance are high [40,45,46,47,48,49,50]. HIF-1a expression is high in neoplasia, and it is responsible for the suppression of tumor cells by inducing NO synthesis, CD 47, PD-L1, and HLA-G overexpression in the tumor microenvironment, overregulation of ADAM 10 metalloproteinase expression, and adenosine increases [20,46,50]. As a result, specific inhibition of HIFs might be useful for improving the antitumor response.

NO could mediate the invasive and metastatic potential of melanoma [51]. In our study, the hypoxia-NO interface was remarkable. In hypoxic conditions, the serum levels of the direct nitrite, total nitrite, and nitrate were significantly higher in the melanoma patients compared with the control group. The serum nitric oxide metabolization dynamic was influenced by the LDH levels, Clark level, Breslow index, tumor stage, and HIF1a and HIF2a levels. Recent studies evaluated the antitumor capacity of NO in the cell lines of melanoma (A2058 and MEL-JUSO) by using an ER selective modulator capable of releasing NO (NO-SERM 4d) [51]. In vitro melanoma cell exposure to NO resulted in PTEN tumor suppression inactivation and RAS-RAF-MEK-ERK NO oncogene signaling promotion. These data support the multiple functions of NO in cancer biology and its role in post-translational changes in tumor suppression proteins [17,45,46,52,53]. A number of processes determine the increased NO levels during tumor hypoxia, and they are as follows: an NOS activity increase, a decrease in the hem oxygenase- or pterine-dependent conversion of NO_2_- to NO, NO tissue deposit release, and cytochrome C mitochondrial oxidase activity modulation [17,54,55,56,57]. An altered energetic metabolism is controlled by NO-induced abnormal proliferation of cancer cells [39,55,58].

Of the many signaling mechanisms of NO, it is appreciated that epigenetic regulation has gained ground in recent years. NO promotes an immunosuppressive microenvironment by modifying the redox state of cells, reprogramming cellular metabolism, and inducing changes in nucleic acids, lipids, and proteins [29,59,60,61,62,63]. All these effects of NO in melanoma are also supported by our results. The present study revealed the amplification of the protein nitration process (3-nitrotyrosine), overexpression of protein (carbonyl group) and lipid (4-hydroxynonenal, malondialdehyde, and reactive substances with thiobarbituric acid) peroxidation, disruption of the thiol-disulfide balance (native thiol, total thiol, and disulfide), and acceleration of the arginine included in protein methylation (symmetrical dimethyl arginine, and asymmetric dimethyl arginine) in patients with melanoma versus a control group.

Under our conditions, the extent of these post-translational changes was influenced by the level of tumor hypoxia and NO production. At the same time, our data showed that oxidative stress occurred early in melanocyte carcinogenesis and was amplified by the disease’s progression. An interesting point of our study was the high levels of ADMA, an endogenous inhibitor of NOS in melanoma. One recent study showed that under conditions of oxidative stress, the synthesis of methyltransferase 1 and the inactivation of DDAH were stimulated, and the levels of ADMA increased [29].

The ROS and NO triggered a signaling cascade that could be transduced by post-translational changes based on oxidation-reduction reactions of the protein thiols. In the melanoma patients in the present study, the increased expression of NO was linked with low prognosis markers of melanoma (advanced Clark levels and Breslow indexes). Much research has shown a correlation between NO and tumor progression by promoting melanogenesis and stimulating pro-tumorigenic cytokines and tumor-associated macrophages [9,10,38,52,63]. On the other hand, NO could limit the proliferation of melanoma by the activation of oncogenes or inactivation of tumor suppressors [64]. Post-translational changes induced by NO and ROS could play a central role in the behavior of cancer cells [37,38,48]. However, the potential role of NO in the progression and treatment of cutaneous melanoma is still under investigation.

Some studies showed that NO could be pivotal in cancer therapy by mediating treatment resistance to cisplatin treatment in melanoma cells in vitro [65]. Additionally, a number of NO-based cancer therapies were developed in order to activate and promote NO synthesis from NOS, to donate NO compounds, and to modulate post-translational protein modifications through S-Nitrosylation and therapies designed to affect or prolong downstream signaling pathways from NO [66].

In summation, the present study is the first one in the literature that evaluates the hypoxia-nitric oxide axis in patients with cutaneous melanoma. Moreover, this study is the sole one focused on monitoring hypoxia and the dynamics of nitric oxide (NO) in different tumoral stages of melanoma. However, some limitations should be noted. Our study followed patients with cutaneous melanoma and with different skin phototypes before receiving any treatment for their disease. Further studies with a larger number of patients with similar skin phototypes and with different therapeutic regimens should be developed. For a better description of the different factors possible for therapeutic targets, patients should be followed up with for a long period of time, and more data according to cancer progression should be collected.

## 5. Conclusions

In melanoma, dysregulation of the NO cycle is an early event that can reach abnormal levels, generating nitrosative-oxidative stress and the disruption of numerous metabolic pathways. The aberrant activation of NO signaling induced by the hypoxic microenvironment was associated with tumor progression. Hypoxia-NO mediated the versatile biological responses dependent on the concentration and duration of exposure to NO, the sensitivity of the molecules with which the NO metabolites interacted, and the redox environment. Our study presented several molecules involved in hypoxia-NO-mediated events that could become potential targets in melanoma treatment in the future.

## Figures and Tables

**Table 1 jpm-12-01646-t001:** Group characteristics.

Clinical Characteristics	Melanoma	Control
	(52 Subjects)	(35 Subjects)
Female/male	29/23	19/16
Age (years)	51.6 ± 10.8	50.8 ± 8.5
BMI (kg/m^2^)	23.9 ± 1.9	22.0 ± 2.4
Systolic pressure (mmHg)	121 ± 19	114 ± 14
Diastolic pressure (mmHg)	60 ± 14	62 ± 13
Skin phototypes I-II/III-IV	32/20	20/15
LDH (U/L)	392.50 ± 35.70	197.31 ± 7.90
Tumor characteristics		
Tumor localization head-neck/trunk/limbs	9/22/21	-
Histopathological type nodular/extensive in surface/lenticular /acral	16/21/7/8	-
Breslow index (mm) <1.0/1.01–2.0/2.01–3.0/>3.01	14/17/12/9	-
Clark level II/III/IV/V	12/17/12/11	-
Lesion ulceration	13/39	-
TNM stage 0/I/II/III/IV	5/7/18/16/6	-

BMI = body mass index; LDH = lactate dehydrogenase.

**Table 2 jpm-12-01646-t002:** HIF-1a and HIF-2a levels in studied groups.

Parameters	Melanoma (52 Subjects)	Control (35 Subjects)	* p * Value
HIF1a (ng/mL)	89.90 ± 32.08	57.63 ± 5.63	0.004
HIF2a (ng/mL)	3.25 ± 0.41	1.40 ± 0.93	0.011
HIF1a/HIF2a	27.31 ± 7.99	41.15 ± 4.07	0.001

HIF-1a = hypoxia-inducible factor–1a; HIF-2a = hypoxia-inducible factor-2a.

**Table 3 jpm-12-01646-t003:** Serum nitric oxide metabolites in studied groups.

Parameters	Melanoma (52 Subjects)	Control (35 Subjects)	* p * Value
Direct nitrite (umols/L)	34.1 ± 6.5	15.2 ± 3.3	0.002
Total nitrite (umols/L)	79.5 ± 11.2	34.0 ± 6.2	0.001
Nitrate (umols/L)	45.4 ± 6.2	18.8 ± 4.6	0.001

**Table 4 jpm-12-01646-t004:** Nitration, carbonylation, thiol-disulphide homeostasis, and methylated arginine in studied groups.

Parameters	Melanoma (52 Subjects)	Control (35 Subjects)	* p * Value
Nitration			
Nitrotyrosine (umol/L)	0.38 ± 0.04	0.13 ± 0.02	˂0.001
Carbonylation			
PCO (μmol/L)	37.82 ± 5.14	22.51 ± 2.21	˂0.001
4-HNE (μgl/mL)	21.15 ± 7.62	14.05 ± 1.39	˂0.001
TBARS (μmol/L)	3.24 ± 0.41	1.97 ± 0.13	˂0.001
MDA (ng/mL)	36.07 ± 5.40	20.08 ± 1.32	˂0.001
Thiol-disulphide homeostasis			
NT (μmol/L)	355.92 ± 8.53	401.83 ± 4.89	˂0.001
TT (μmol/L)	407.23 ± 7.40	440.89 ± 4.78	˂0.001
DS (μmol/L)	25.65 ± 1.62	19.50 ± 0.53	˂0.001
DS/NT	7.21 ± 0.55	4.85 ± 0.15	˂0.001
DS/TT	6.30 ± 0.42	3.98 ±1.27	˂0.001
NT/TT	87.39 ± 0.85	91.15 ± 0.25	˂0.001
Methylated arginine			
SDMA (μmol/L)	0.96 ± 0.10	0.50 ± 0.03	˂0.001
ADMA (μmol/L)	0.86 ± 0.09	0.58 ± 0.05	˂0.001
ADMA/SDMA	0.90 ± 0.07	1.10 ± 0.10	˂0.001

PCO = carbonylated proteins; 4-HNE = 4-hydroxy-2-nonenal; MDA = malondialdehyde; TBARS = thiobarbituric acid-reactive substances; NT = native thiol; TT = total thiol; DS = disulfide; SDMA = symmetric dimethylarginine; ADMA = asymmetric dimethylarginine.

**Table 5 jpm-12-01646-t005:** HIF-1a and HIF-2a levels in melanoma according to Breslow index.

Parameters	Breslow Index						
	<1.0	1.01–2.0		2.01–3.0		>3.01	
HIF-1a (ng/mL)	63.90 ± 22.70	84.90 ± 28.31	*p*1 < 0.05	98.35 ± 33.66	*p*1 < 0.05 *p*2 < 0.05	112.32 ± 43.69	*p*1 < 0.05 *p*2 < 0.05 *p*3 < 0.05
HIF-2a (ng/mL)	2.87 ± 0.26	3.11 ± 0.34	*p*1 < 0.05	3.41 ± 0.47	*p*1 < 0.05 *p*2 < 0.05	3.62 ± 0.58	*p*1 < 0.05 *p*2 < 0.05 *p*3 < 0.05
HIF-1a/HIF-2a	35.61 ± 10.37	28.22 ± 8.23	*p*1 < 0.05	26.12 ± 7.77	*p*1 < 0.05 *p*2 > 0.05	19.3 ± 5.62	*p*1 < 0.05 *p*2 < 0.05 *p*3 < 0.05

HIF-1a = hypoxia-inducible factor-1a; HIF-2a = hypoxia-inducible factor-2a; *p*1 = Breslow index (1.01–2.0, 2.01–3.0, and >3.01 vs. <1.0); *p*2 = Breslow index (2.01–3.0 and >3.01 vs. 1.01–2.0), *p*3 = Breslow index (>3.01 vs. 2.01–3.0).

**Table 6 jpm-12-01646-t006:** HIF-1a and HIF-2a levels in melanoma according to Clark level.

Parameters	Clark Level						
	II	III		IV		V	
HIF-1a (ng/mL)	61.34 ± 17.21	86.74 ± 26.83	*p*1 < 0.05	99.66 ± 37.98	*p*1 < 0.05 *p*2 < 0.05	111.62 ± 46.03	*p*1 < 0.05 *p*2 < 0.05 *p*3 < 0.05
HIF-2a (ng/mL)	2.68 ± 0.28	3.22 ± 0.35	*p*1 < 0.05	3.45 ± 0.42	*p*1 < 0.05 *p*2 ˃ 0.05	3.67 ± 0.59	*p*1 < 0.05 *p*2 < 0.05 *p*3 < 0.05
HIF-1a/HIF-2a	35.82 ± 8.86	28.12 ± 8.40	*p*1 < 0.05	26.41 ± 8.02	*p*1 < 0.05 *p*2 > 0.05	18.9 ± 6.69	*p*1 < 0.05 *p*2 < 0.05 *p*3 < 0.05

HIF-1a = hypoxia-inducible factor=1a; HIF-2a = hypoxia-inducible factor-2a; *p*1 = Clark level (III, IV, and V vs. II); *p*2 = Clark level (IV and V vs. III); *p*3 = Clark level (V vs. IV).

**Table 7 jpm-12-01646-t007:** HIF-1a and HIF-2a levels according to melanoma stage.

Parameters	Melanoma Stage						
	I	II		III		IV	
HIF-1a (ng/mL)	61.30 ± 18.21	86.78 ± 27.43	*p*1 < 0.05	99.67 ± 37.16	*p*1 < 0.05 *p*2 < 0.05	111.82 ± 45.53	*p*1 < 0.05 *p*2 < 0.05 *p*3 < 0.05
HIF-2a (ng/mL)	2.62 ± 0.31	3.14 ± 0.35	*p*1 < 0.05	3.44 ± 0.39	*p*1 < 0.05 *p*2 > 0.05	3.72 ± 0.61	*p*1 < 0.05 *p*2 < 0.05 *p*3 > 0.05
HIF-1a/HIF-2a	36.20 ± 8.82	26.87 ± 8.73	*p*1 < 0.05	27.38 ± 7.87	*p*1 < 0.05 *p*2 > 0.05	18.81 ± 6.55	*p*1 < 0.05 *p*2 < 0.05 *p*3 < 0.05

HIF-1a = hypoxia-inducible factor-1a; HIF-2a = hypoxia-inducible factor-2a; *p*1 = melanoma stage (II, III, and IV vs. I), *p*2 = melanoma stage (III and IV vs. II); *p*3 = melanoma stage (IV vs. III).

**Table 8 jpm-12-01646-t008:** Serum nitric oxide metabolites in melanoma according to Breslow index.

Parameters	Breslow Index						
	<1.0	1.01–2.0		2.01–3.0		>3.01	
Direct nitrite (umols/L)	28.12 ± 4.38	31.75 ± 6.88	*p*1 < 0.05	36.34 ± 6.88	*p*1 < 0.05 *p*2 < 0.05	40.42 ± 8.47	*p*1 < 0.05 *p*2 < 0.05 *p*3 < 0.05
Total nitrite (umols/L)	70.52 ± 8.33	76.83 ± 10.62	*p*1 < 0.05	80.9 ± 10.62	*p*1 < 0.05 *p*2 < 0.05	89.77 ± 13.25	*p*1 < 0.05 *p*2 < 0.05 *p*3 < 0.05
Nitrate (umols/L)	38.21 ± 4.77	39.73 ± 5.39	*p*1 > 0.05	41.59 ± 6.52	*p*1 < 0.05 *p*2 > 0.05	62.08 ± 8.11	*p*1 < 0.05 *p*2 < 0.05 *p*3 < 0.05

*p*1 = Breslow index 1.01–2.0, 2.01–3.0, and >3.01 vs. <1.0; *p*2 = Breslow index 2.01–3.0 and >3.01 vs. 1.01–2.0; *p*3 = Breslow index >3.01 vs. 2.01–3.0.

**Table 9 jpm-12-01646-t009:** Serum nitric oxide metabolites in melanoma according to Clark level.

Parameters	Clark Level						
	II	III		IV		V	
Direct nitrite (umols/L)	28.56 ± 4.22	30.72 ± 5.62	*p*1 < 0.05	36.45 ± 7.34	*p*1 < 0.05 *p*2 < 0.05	40.79 ± 8.96	*p*1 < 0.05 *p*2 < 0.05 *p*3 < 0.05
Total nitrite (umols/L)	71.38 ± 8,73	74.67 ± 10.34	*p*1 < 0.05	81.49 ± 11.68	*p*1 < 0.05 *p*2 < 0.05	90.52 ± 14.05	*p*1 < 0.05 *p*2 < 0.05 *p*3 < 0.05
Nitrate (umols/L)	37.65 ± 4.82	39.23 ± 5.71	*p*1 > 0.05	42.31 ± 6.02	*p*1 < 0.05 *p*2 < 0.05	62.42 ± 8.27	*p*1 < 0.05 *p*2 < 0.05 *p*3 < 0.05

*p*1 = Clark levels III, IV, and V vs. II; *p*2 = Clark levels IV and V vs. III; *p*3 = Clark level V vs. IV.

**Table 10 jpm-12-01646-t010:** Serum nitric oxide metabolites according to melanoma stage.

Parameters	Melanoma Stage						
	I	II		III		IV	
Direct nitrite (umols/L)	29.83 ± 4.57	30.72 ± 5.69	*p*1 > 0.05	35.27 ± 7.78	*p*1 < 0.05 *p*2 < 0.05	41.04 ± 7.93	*p*1 < 0.05 *p*2 < 0.05 *p*3 < 0.05
Total nitrite (umols/L)	71.92 ± 9.02	75.15 ± 10.77	*p*1 > 0.05	81.79 ± 11.21	*p*1 < 0.05 *p*2 > 0.05	89.17 ± 13.82	*p*1 < 0.05 *p*2 < 0.05 *p*3 < 0.05
Nitrate (umols/L)	37.19 ± 5.02	38.73 ± 5.51	*p*1 > 0.05	44.21 ± 6.12	*p*1 < 0.05 *p*2 < 0.05	61.69 ± 8.17	*p*1 < 0.05 *p*2 < 0.05 *p*3 < 0.05

*p*1 = melanoma stages II, III, and IV vs. I; *p*2 = melanoma stages III and IV vs. II; *p*3 = melanoma stage IV vs. III.

**Table 11 jpm-12-01646-t011:** Nitration and carbonylation in melanoma according to Breslow index.

Parameters	Breslow Index						
	<1.0	1.01–2.0		2.01–3.0		>3.01	
Nitrotyrosine (umol/L)	0.33 ± 0.03	0.36 ± 0.02	*p*1 < 0.05	0.41 ± 0.02	*p*1 < 0.05 *p*2 > 0.05	0.40 ± 0.02	*p*1 < 0.05 *p*2 < 0.05 *p*3 > 0.05
Carbonylation							
PCO (μmol/L)	34.28 ± 2.26	36.84 ± 5.86	*p*1 < 0.05	40.85 ± 4.80	*p*1 < 0.05 *p*2 < 0.05	43.01 ± 2.73	*p*1 < 0.05 *p*2 < 0.05 *p*3 < 0.05
4-HNE (μgl/mL)	18.44 ± 6.82	21.26 ± 7.92	*p*1 < 0.05	21.84 ± 7.70	*p*1 < 0.05 *p*2 > 0.05	23.08 ± 8.05	*p*1 < 0.05 *p*2 < 0.05 *p*3 > 0.05
TBARS (μmol/L)	3.06 ± 0.36	3.27 ± 0.42	*p*1 < 0.05	3.18 ± 0. 36	*p*1 < 0.05 *p*2 < 0.05	3.74 ± 0.08	*p*1 < 0.05 *p*2 < 0.05 *p*3 < 0.05
MDA (ng/mL)	31.92 ± 3.51	35.73 ± 6.19	*p*1 < 0.05	38.85 ± 3.37	*p*1 < 0.05 *p*2 < 0.05	41.20 ±1.09	*p*1 < 0.05 *p*2 < 0.05 *p*3 < 0.05

PCO = carbonylated proteins; 4-HNE = 4-hydroxy-2-nonenal; MDA = malondialdehyde; TBARS = thiobarbituric acid-reactive substances; *p*1 = Breslow indexes 1.01–2.0, 2.01–3.0, and >3.01 vs. <1.0; *p*2 = Breslow indexes 2.01–3.0 and >3.01 vs. 1.01–2.0; *p*3 = Breslow index >3.01 vs. 2.01–3.0.

**Table 12 jpm-12-01646-t012:** Nitration and carbonylation in melanoma patients according to Clark level.

Parameters	Clark Level						
	II	III		IV		V	
Nitration							
Nitrotyrosine (umol/L)	0.35 ± 0.03	0.37 ± 0.04	*p*1 < 0.05	0.40 ± 0.08	*p*1 < 0.05 *p*2 < 0.05	0.39 ± 0.03	*p*1 < 0.05 *p*2 < 0.05 *p*3 < 0.05
Carbonylation							
PCO (μmol/L)	33.26 ± 4.19	36.67 ± 4.28	*p*1 < 0.05	39.52 ± 5.08	*p*1 < 0.05 *p*2 < 0.05	41.85 ± 7.02	*p*1 < 0.05 *p*2 < 0.05 *p*3 < 0.05
4-HNE (μgl/mL)	19.44 ± 7.12	21.26 ± 7.90	*p*1 < 0.05	21.60 ± 7.54	*p*1 < 0.05 *p*2 > 0.05	22.31 ± 7.95	*p*1 > 0.05 *p*2 > 0.05 *p*3 < 0.05
TBARS (μmol/L)	3.10 ± 0.44	2.94 ± 0.08	*p*1 > 0.05	3.67 ± 0.22	*p*1 < 0.05 *p*2 < 0.05	3.49 ± 0.40	*p*1 < 0.05 *p*2 < 0.05 *p*3 > 0.05
MDA (ng/mL)	32.33 ± 5.65	33.66 ± 3.01	*p*1 > 0.05	39.60 ± 5.69	*p*1 < 0.05 *p*2 < 0.05	40.51 ± 2.50	*p*1 < 0.05 *p*2 < 0.05 *p*3 < 0.05

PCO = carbonylated proteins; 4-HNE = 4-hydroxy-2-nonenal; MDA = malondialdehyde; TBARS = thiobarbituric acid-reactive substances; *p*1 = Clark levels III, IV, and V vs. II; *p*2 = Clark levels IV and V vs. III; *p*3 = Clark level V vs. IV.

**Table 13 jpm-12-01646-t013:** Nitration and carbonylation according to melanoma stage.

Parameters	Melanoma Stage						
	I	II		III		IV	
Nitration							
Nitrotyrosine (umol/L)	0.35 ± 0.04	0.37 ± 0.03	*p*1 > 0.05	0.39 ± 0.06	*p*1 < 0.05 *p*2 < 0.05	0.42 ± 0.05	*p*1 < 0.05 *p*2 < 0.05 *p*3 < 0.05
Carbonylation							
PCO (μmol/L)	32.91 ± 3.45	35.84 ± 4.37	*p*1 < 0.05	39.68 ± 5.08	*p*1 < 0.05 *p*2 < 0.05	42.86 ± 7.69	*p*1 < 0.05 *p*2 < 0.05 *p*3 < 0.05
4-HNE (μgl/mL)	19.85 ± 6.89	20.67 ± 8.01	*p*1 > 0.05	21.49 ± 7.94	*p*1 < 0.05 *p*2 > 0.05	22.62 ± 7.65	*p*1 < 0.05 *p*2 < 0.05 *p*3 > 0.05
TBARS (μmol/L)	3.06 ± 0.37	3.27 ± 0.42	*p*1 < 0.05	3.15 ± 0.34	*p*1 < 0.05 *p*2 > 0.05	3.61 ± 0.32	*p*1 < 0.05 *p*2 < 0.05 *p*3 < 0.05
MDA (ng/mL)	31.46 ± 3.51	36.01 ± 6.5	*p*1 < 0.05	38.06 ± 3.76	*p*1 < 0.05 *p*2 < 0.05	41.01 ± 1.09	*p*1 < 0.05 *p*2 < 0.05 *p*3 < 0.05

PCO = carbonylated proteins; 4-HNE = 4-hydroxy-2-nonenal; MDA = malondialdehyde, TBARS = thiobarbituric acid-reactive substances; *p*1 = melanoma stages II, III, and IV vs. I; *p*2 = melanoma stages III and IV vs. II; *p*3 = melanoma stage IV vs. III.

**Table 14 jpm-12-01646-t014:** Serum nitric oxide metabolites in melanoma according to Breslow index.

Parameters	Breslow Index						
	<1.0	1.01–2.0		2.01–3.0		>3.01	
SDMA (μmol/L)	0.88 ± 0.07	0.93 ± 0.07	*p*1 < 0.05	0.97 ± 0.08	*p*1 < 0.05 *p*2 < 0.05	1.04 ± 0.07	*p*1 < 0.05 *p*2 < 0.05 *p*3 < 0.05
ADMA (μmol/L)	0.78 ± 0.07	0.84 ± 0.09	*p*1 < 0.05	0.90 ± 0.09	*p*1 < 0.05 *p*2 < 0.05	0.94 ± 0.11	*p*1 < 0.05 *p*2 < 0.05 *p*3 < 0.05
ADMA/SDMA	0.88 ± 0.05	0.90 ± 0.06	*p*1 > 0.05	0.92 ± 0.08	*p*1 > 0.05 *p*2 > 0.05	0.90 ± 0.09	*p*1 > 0.05 *p*2 > 0.05 *p*3 > 0.05

SDMA = symmetric dimethylarginine; ADMA = asymmetric dimethylarginine; *p*1 = Breslow indexes 1.01–2.0, 2.01–3.0, and >3.01 vs. <1.0; *p*2 = Breslow indexes 2.01–3.0 and >3.01 vs. 1.01–2.0; *p*3 = Breslow index >3.01 vs. 2.01–3.0.

**Table 15 jpm-12-01646-t015:** Methylated arginine metabolites in melanoma according to Clark level.

Parameters	Clark Level						
	II	III		IV		V	
SDMA (μmol/L)	0.91 ± 0.08	0.92 ± 0.04	*p*1 > 0.05	0.97 ± 0.05	*p*1 < 0.05 *p*2 < 0.05	1.02 ± 0.05	*p*1 < 0.05 *p*2 < 0.05 *p*3 > 0.05
ADMA (μmol/L)	0.79 ± 0.07	0.83 ± 0.08	*p*1 < 0.05	0.91 ± 0.1	*p*1 < 0.05 *p*2 < 0.05	0.94 ± 0.12	*p*1 < 0.05 *p*2 < 0.05 *p*3 < 0.05
ADMA/SDMA	0.88 ± 0.06	0.90 ± 0.05	*p*1 > 0.05	0.94 ± 0.07	*p*1 > 0.05 *p*2 > 0.05	0.87 ± 0.07	*p*1 > 0.05 *p*2 > 0.05 *p*3 < 0.05

SDMA = symmetric dimethylarginine; ADMA = asymmetric dimethylarginine; *p*1 = Clark levels III, IV, and V vs. II; *p*2 = Clark levels IV and V vs. III; *p*3 = Clark level V vs. IV.

**Table 16 jpm-12-01646-t016:** Methylated arginine according to melanoma stage.

Parameters	Melanoma Stage						
	I	II		III		IV	
SDMA (μmol/L)	0.90 ± 0.07	0.93 ± 0.05	*p*1 < 0.05	0.97 ± 0.04	*p*1 < 0.05 *p*2 < 0.05	1.02 ± 0.07	*p*1 < 0.05 *p*2 < 0.05 *p*3 < 0.05
ADMA (μmol/L)	0.77 ± 0.06	0.83 ± 0.08	*p*1 < 0.05	0.91 ± 0.10	*p*1 < 0.05 *p*2 < 0.05	0.95 ± 0.12	*p*1 < 0.05 *p*2 < 0.05 *p*3 < 0.05
ADMA/SDMA	0.88 ± 0.05	0.88 ± 0.07	*p*1 > 0.05	0.93 ± 0.09	*p*1 > 0.05 *p*2 > 0.05	0.89 ± 0.09	*p*1 > 0.05 *p*2 > 0.05 *p*3 < 0.05

SDMA = symmetric dimethylarginine; ADMA = asymmetric dimethylarginine; *p*1 = melanoma stages II, III, and IV vs. I; *p*2 = melanoma stages III and IV vs. II; *p*3 = melanoma stage IV vs. III.

**Table 17 jpm-12-01646-t017:** The interdependence between microenvironment soluble factors and tumor characteristics.

Parameters		LDH	Clark	Breslow	TNM	Ulceration
HIF-1a	*r*	0.250	0.690	0.400	0.410	0.010
	*p*	0.061	0.020	0.031	0.055	0.890
HIF-2a	*r*	0.554	0.192	0.206	0.484	0.040
	*p*	0.002	0.150	0.154	0.041	0.895
HIF-1a/ HIF-2a	*r*	0.257	−0.189	−0.200	−0.250	−0.030
	*p*	0.070	0.050	0.002	0.040	0.760
Direct nitrite	*r*	0.401	0.630	0.390	0.350	0.120
	*p*	0.030	0.001	0.020	0.040	0.290
Nitrate	*r*	0.380	0.580	0.460	0.460	−0.020
	*p*	0.005	0.010	0.001	0.001	0.780
3-nitrotirosine	*r*	0.290	0.290	0.360	0.190	0.140
	*p*	0.030	0.010	0.009	0.170	0.200
Carbonylic groups	*r*	0.280	0.640	0.419	0.190	0.140
	*p*	0.040	0.010	0.001	0.150	0.200
4-HNE	*r*	0.400	0.600	0.420	0.350	0.030
	*p*	0.003	0.010	0.001	0.005	0.790
TBARS	*r*	0.450	0.050	0.306	0.200	0.180
	*p*	0.001	0.001	0.020	0.130	0.150
MDA	*r*	0.240	0.620	0.470	0.300	0.070
	*p*	0.080	0.010	0.001	0.010	0.500
NT	*r*	−0.381	0.420	−0.450	−0.110	0.040
	*p*	0.009	0.020	0.001	0.400	0.730
TT	*r*	−0.331	−0.450	−0.480	0.010	0.040
	*p*	0.010	0.010	0.001	0.950	0.710
DS	*r*	0.170	0.080	0.800	0.270	0.030
	*p*	0.200	0.550	0.580	0.600	0.950
DS/NT	*r*	0.250	0.190	0.200	0.250	0.010
	*p*	0.061	0.160	0.151	0.080	0.890
DS/TT	*r*	0.254	0.192	0.206	0.260	0.040
	*p*	0.062	0.150	0.154	0.081	0.895
NT/TT	*r*	0.257	0.189	−0.200	−0.250	0.030
	*p*	0.070	0.150	0.152	0.080	0.760
SDMA	*r*	0.010	0.570	−0.280	0.180	0.106
	*p*	0.430	0.010	0.050	0.200	0.010
ADMA	*r*	0.340	0.640	0.284	0.284	0.040
	*p*	0.010	0.010	0.030	0.010	0.950
ADMA/SDMA	*p*	0.156	0.280	0.100	0.230	0.120
	*r*	0.050	0.030	0.480	0.120	0.754

HIF-1a = hypoxia-inducible factor-1a; HIF-2a = hypoxia-inducible factor-2a; PCO = carbonylated proteins; 4-HNE = 4-hydroxy-2-nonenal; MDA = malondialdehyde; TBARS = thiobarbituric acid-reactive substances; NT = native thiol; TT = total thiol; DS = disulfide; SDMA = symmetric dimethylarginine; ADMS = asymmetric dimethylarginine; *r* = correlation coefficient; *p* = statistical significance.

**Table 18 jpm-12-01646-t018:** The interdependence between HIF-NO axis and damage-related molecular pattern in melanoma group.

Metabolites		HIF1a	HIF2a	HIF1a/HIF2a	Direct Nitrite	Nitrate
Direct nitrite	*r*	0.654	0.331	0.342	-	-
	*p*	0.018	0.036	0.004	-	-
Nitrate	*r*	0.730	0.470	0.375	-	-
	*p*	0.020	0.006	0.030	-	-
3-nitrotirosine	*r*	0.311	0.490	0.578	0.870	0.860
	*p*	0.028	0.041	0.001	0.001	0.002
Carbonylic groups	*r*	0.401	0.720	0.653	0.520	0.430
	*p*	0.018	0.006	0.001	0.001	0.001
4-HNE	*r*	0.392	0.822	0.720	0.510	0.470
	*p*	0.002	0.001	0.001	0.001	0.008
TBARS	*r*	0.412	0.630	0.750	0.376	0.609
	*p*	0.001	0.001	0.001	0.022	0.010
MDA	*r*	0.371	0.520	0.540	0.510	0.420
	*p*	0.002	0.010	0.002	0.002	0.040
NT	*r*	−0.530	−0.491	0.320	−0.510	−0.150
	*p*	0.001	0.004	0.020	0.001	0.260
TT	*r*	−0.553	−0.367	0.450	−0.422	−0.110
	*p*	0.002	0.038	0.020	0.003	0.650
DS	*r*	0.287	0.281	0.140	0.768	0.297
	*p*	0.043	0.002	0.420	0.040	0.030
DS/NT	*r*	−0.301	−0.250	0.090	0.130	0.310
	*p*	0.015	0.072	0.470	0.201	0.140
DS/TT	*r*	−0.404	−0.221	0.212	0.176	0.130
	*p*	0.041	0.059	0.320	0.177	0.111
NT/TT	*r*	0.398	0.306	0.169	0.160	−0.340
	*p*	0.051	0.040	0.350	0.432	0.020
SDMA	*r*	0.110	0.146	0.210	0.195	0.211
	*p*	0.010	0.176	0.210	0.360	0.342
ADMA	*r*	0.490	0.587	0.730	0.684	0.810
	*p*	0.031	0.028	0.002	0.001	0.001
ADMA/SDMA	*r*	−0.098	0.203	0.320	0.126	0.260
	*p*	0.520	0.043	0.033	0.218	0.430

HIF-1a = hypoxia-inducible factor-1a; HIF-2a = hypoxia-inducible factor-2a; PCO = carbonylated proteins; 4-HNE = 4-hydroxy-2-nonenal; MDA = malondialdehyde; TBARS = thiobarbituric acid-reactive substances; NT = native thiol; TT = total thiol; DS = disulfide; SDMA = symmetric dimethylarginine; ADMS = asymmetric dimethylarginine; *r* = correlation coefficient; *p* = statistical significance.

## Data Availability

The data presented in this study are available on request from the corresponding author.

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
