# Peer review of "Hypoxia-Nitric Oxide Axis and the Associated Damage Molecular Pattern in Cutaneous Melanoma"

_jpm, 2022, doi:10.3390/jpm12101646_

Round 1

Reviewer 1 Report

Line 55: TAM - no explanation of the abbreviation

Line 56: MDSC - no explanation of the abbreviation

Line 75: reference

Line 91: reference

Line 91 and Line 373: You wrote about ‘possible strategies that modulate NO homeostasis’ and ‘possible therapeutic target’, are there any possible therapies nowadays? what are these potential therapeutic strategies? Please describe them shortly.

Table 1 : are systolic and diastolic pressure have proper numbers?

Line 368: ‘the present study is the first one in the literature that evaluates hypoxia – nitric oxide axis in patients with cutaneous melanoma in different tumoral stages.’ Does it mean that no studies that evaluates hypoxia – nitric oxide axis in cutaneous melanoma patients were never published or just in melanoma different tumoral stages?

A lot of Discussion section just describes literature data and would be more fitting to place in the introductory section. The Discussion should focus on comparing the results with literature data.

Author Response

Dear Reviewer,

Thank you for giving me the opportunity to submit a revised draft of my manuscript titled

Hypoxia – Nitric Oxide Axis and Associated Damage Molecular Pattern in Cutaneous Melanoma to your Journal. We appreciate the time and effort that you and the reviewers have dedicated to providing your valuable feedback on the manuscript. We are grateful to the reviewers for their insightful comments on the paper. We have been able to incorporate changes to reflect most of the suggestions provided by the reviewers.

Here is a point-by-point response to the reviewers’ comments and concerns.

  1. Line 55: TAM - no explanation of the abbreviation

We added the explanation - Tumor-associated macrophages,

  1. Line 56: MDSC - no explanation of the abbreviation-

We added the explanation - Myeloid-derived suppressor cells.

  1. Line 75: reference

We added the reference no [31].

  1. Line 91: reference

We changed the formulation if the sentence: The results of the present study

  1. Line 91 and Line 373: You wrote about ‘possible strategies that modulate NO homeostasis’ and ‘possible therapeutic target’, are there any possible therapies nowadays? what are these potential therapeutic strategies? Please describe them shortly.

We described the NO possible therapeutic target and any therapies in development nowadays in the end of the Discussions section.

  1. Table 1 : are systolic and diastolic pressure have proper numbers?

No.We have changed with proper numbers.

  1. Line 368: ‘the present study is the first one in the literature that evaluates hypoxia – nitric oxide axis in patients with cutaneous melanoma in different tumoral stages.’ Does it mean that no studies that evaluates hypoxia – nitric oxide axis in cutaneous melanoma patients were never published or just in melanoma different tumoral stages?

The present study is the first study in literature that evaluated hypoxia – nitric oxide axis in cutaneous melanoma. Analyzing this axis in all melanoma stages is a key point in our study.

  1. A lot of Discussion section just describes literature data and would be more fitting to place in the introductory section. The Discussion should focus on comparing the results with literature data.

The Discussion section was rearranged focusing on our results in relation with data in literature

Reviewer 2 Report

Dear Authors,

I would like to appreciate your work and the idea. Melanoma of any subtype is a relevant issue and needs to be studied more. Even I like the idea a have some significant comments.

First of all, the whole article has grammar mistakes and the sentences are hard to read. Some examples - line 17: hypoxia-inducible factors; line 19: oxide synthases; tumor or tumour?; indexes (upper/downer) in the chemical formulas; space in front of and behind "/", etc.

Please add some reference to the sentence in lines 41/42.

Explain abbreviations such as TAM (line 55), MDSC (line 56), etc.

Equation (1) and (2) is in the text named "(a)" and "(b)" (line 59, line 64).

Line 68/69 - NORMOXIA

Line 126 - cell lysates? I thought that you used blood serum...

Line 147/148 - unify the units (abbreviation or?)

Please mention in the abstract and introduction that protein expression was analyzed.

Line 191/192 - skin phototype is a well-known factor of melanoma presence... For an effective analysis, you need the same condition, which also means the same skin phototype...

Results - please show the results divided by tumor classification (stage, size, etc.) not only cancer vs. non-cancer - it'll show up more information (it is highly needed for each group of results shown in table 2,3,4...).

I don't understand what numbers 3 and 4 mean - line 252. 3 Nitrothzrosine...; line 254: 4 HNE...

Line 305: the sentence Other studies showed that ... - also other studies clarified that HIF1a acts as a tumor stimulant...

The discussion seems to be more introduction than the discussion itself. It needs to improve.

Author Response

Dear Reviewer,

Thank you for giving me the opportunity to submit a revised draft of my manuscript titled

Hypoxia – Nitric Oxide Axis and Associated Damage Molecular Pattern in Cutaneous Melanoma to your Journal. We appreciate the time and effort that you and the reviewers have dedicated to providing your valuable feedback on the manuscript. We are grateful to the reviewers for their insightful comments on the paper. We have been able to incorporate changes to reflect most of the suggestions provided by the reviewers.

Here is a point-by-point response to the reviewers’ comments and concerns.

  1. First of all, the whole article has grammar mistakes and the sentences are hard to read. Some examples - line 17: hypoxia-inducible factors; line 19: oxide synthases; tumor or tumour?; indexes (upper/downer) in the chemical formulas; space in front of and behind "/", etc.

We corrected the grammar mistakes and rearranged the senteces.

  1. Please add some reference to the sentence in lines 41/42.

We added the reference no [11].

  1. Explain abbreviations such as TAM (line 55), MDSC (line 56), etc.

We explained the abbreviation, as follows:  Tumor-associated macrophages, Myeloid-derived suppressor cells

  1. Equation (1) and (2) is in the text named "(a)" and "(b)" (line 59, line 64).

We made the changes in the rext.

  1. Line 68/69 – NORMOXIA

We made the changes in the text.

  1. Line 126 - cell lysates? I thought that you used blood serum...

We used subjects’ serum. We modified the typing error.

  1. Line 147/148 - unify the units (abbreviation or?)

We made the changes in the text.

  1. Line 191/192 - skin phototype is a well-known factor of melanoma presence... For an effective analysis, you need the same condition, which also means the same skin phototype...

We mentioned as a study limitation the presence of multiple skin phototypes.

  1. Results - please show the results divided by tumor classification (stage, size, etc.) not only cancer vs. non-cancer - it'll show up more information (it is highly needed for each group of results shown in table 2,3,4...).

We added the tables in Appendix A.

  1. I don't understand what numbers 3 and 4 mean - line 252. 3-Nitrothzrosine...; line 254: 4-HNE... We corrected the typing errors in the text.
  2. The discussion seems to be more introduction thn the discussion itself. It needs to improve.

The Discussion section was rearranged focusing on our results in relation with data in literature.

Round 2

Reviewer 2 Report

Dear Authors,

thank you for your prompt work on the manuscript. Hovewer, I found some topics that have to be done:

line 16 - ...(HIF-1s)oxid... - what is "oxid"?

line 19 - ... (NOs)tu... - what is "tu"?

line 84 - and cy-totoxicity (no "-" needed)

line 83-86 - more references needed

line 264 - 3-Nitrothyrosine - correct to 3-Nitrotyrosine (the same in the table 5)

line 283 - 4 HNE - correct to 4-HNE

table 4 and 5 - what is "r" and "p"? - missing explanation

But more important is that all results given in "Appendix" are more valuable than those in regular Result section. Please, transfer "Appendix" results into Results. All "Appendix" results show only p-values </> 0.05. Please, be more concrete.

Author Response

Dear Reviewer,

Thank you for giving me the opportunity to submit a revised draft of my manuscript titled Hypoxia – Nitric Oxide Axis and Associated Damage Molecular Pattern in Cutaneous Melanoma to your Journal. We appreciate the time and effort that you have dedicated to providing your valuable feedback on the manuscript. We are grateful for your insightful comments on the paper. We have been able to incorporate changes to reflect most of the suggestions provided by you.

Here is a point-by-point response to the reviewers’ comments and concerns.

  1. ine 16 - ...(HIF-1s)oxid... - what is "oxid"? We corrected the typing error.
  2. line 19 - ... (NOs)tu... - what is "tu"? We corrected the typing error.
  3. line 84 -and cy-totoxicity (no "-" needed) We corrected the typing error.
  4. line 83-86 - more references needed We added new redderences.
  5. line 264 - 3-Nitrothyrosine - correct to 3-Nitrotyrosine (the same in the table 5) We corrected the typing error.
  6. line 283 - 4 HNE - correct to 4-HNE We corrected the typing error.
  7. table 4 and 5 - what is "r" and "p"? - missing explanation We completed the missing explanation.
  8. But more important is that all results given in "Appendix" are more valuable than those in regular Result section. Please, transfer "Appendix" results into Results. All "Appendix" results show only p-values </> 0.05. Please, be more concrete. We  transferred "Appendix" results into Results and developed this section.